# The dependence of hydropower planning in relation to the influence of climate in Northeast Brazil

**Nicorray de Queiroz Santos** [ORCID]*, **Kellen Carla Lima**⊗, **Maria Helena Constantino Spyrides**⊗

Postgraduate Program in Climate Science, Center for Exact and Earth Sciences, Federal University of Rio Grande do Norte, Natal, Rio Grande do Norte, Brazil

⊗ These authors contributed equally to this work.
* nicorray.queiroz@gmail.com

**Data Availability Statement:** The data underlying the results presented in the study are available from: ENSO.34: https://www.cpc.ncep.noaa.gov/products/analysis_monitoring/ensostuff/detrend.nino34.ascii.txt O Niño Index: https://origin.cpc.ncep.noaa.gov/products/analysis_monitoring/

## Abstract

Water scarcity in Northeast Brazil has caused latent perturbations in hydropower generation, which is undesirable for Energy Planning. Thus, this study aims to identify the influence of climate on hydropower generation by Sobradinho Dam in Bahia by: (i) assessing the streamflow climatology (1964–2017) and rainfall (1964–2015) through time series analysis, hypothesis testing and cluster analysis; (ii) assessing hydropower generation (2000–2017) using climate and energy data, through principal component analysis and dynamic regression models. The results indicated reductions of up to 30% in the mean climatological streamflow patterns; and reductions in rainfall amounts of 22.9%, 13.3% and 12.4% at latitudes 9°, 12°and 13°South, respectively. Decreasing trends were found in simulations of hydropower generation under the influence of different climate variables. Thus, the hydroelectric system operates in contingency, due to the growing energy load demand resulting in more energy imports in Northeast Brazil.

## Introduction

Northeast Brazil (*NEB*) has experienced a period of severe water shortage, as revealed by the historical series of streamflow and useful volume of the Sobradinho reservoir in the state of Bahia [1–4]. It is difficult to conceive a natural disaster with these characteristics in Brazil, but drought is in fact a recurring hazard in this region [5, 6]. Given that the Brazilian energy mix is mainly reliant on hydrothermal generation, which supplies 63.76% of the nation's demand, climate is of the most importance for the energy sector and for energy planning (*EP*) [7].

Periods of prolonged and persistent drought [6, 8] usually lead to discussions, including of a legal nature, prompting the elaboration of instruments, such as, the Brazilian *Energy Reallocation Mechanism* (*ERM*—mechanism for sharing hydrological risks [9], associated with the optimization of the National Interconnected System—*NIS*). This was created in 1998 by Decree 2,655. Given the impacts of the hydrological crisis caused by water scarcity [2, 10, 11], this decree aimed to apportion the risk faced by hydropower generators, which amplifies commercial issues, especially when the *ERM* is lower than the generated Physical Guarantee

ensostuff/ONI_v5.php Sea surface temperature [south SST]: https://www.cpc.ncep.noaa.gov/data/indices/sstoi.indices Atlantic index: https://www.cpc.ncep.noaa.gov/data/indices/sstoi.atl.indices TSA/TNA: https://www.esrl.noaa.gov/psd/data/climateindices/list/ https://www.esrl.noaa.gov/psd/data/correlation/tna.data https://www.esrl.noaa.gov/psd/data/correlation/tsa.data Energy data - www.ons.org.br INMET - https://bdmep.inmet.gov.br/ GPCP - https://www.esrl.noaa.gov/psd/data/gridded/data.gpcp.html.

**Funding:** The author(s) received no specific funding for this work.

**Competing interests:** The authors have declared that no competing interests exist.

(*PG*—maximum amount in MW of a power plant). However, this risk-sharing model does not seem reasonable given the critical water availability conditions in *NEB*, which is associated with changes in the water cycle, with rising energy demand during most of operational periods, causing the need to import energy from other submarkets (division of the *NIS* for which specific liquidation values are established and with boundaries defined based on the presence and duration of relevant transmission restrictions on power flow) [4, 12, 13].

An important parameter for hydrological risk management is the Generation Scale Factor (*GSF*), which measures the monthly ratio between energy produced by the *ERM* generators and the sum of their *PG* [12, 14]. This mechanism is very sensitive to water scarcity, since it increases the risk of energy supply to the system. This occurs due to probable misconceived assumptions of the initial climate analysis [15]. In the Sinu-Caribe basin in Colombia, for example, there was an 8.5% decrease in streamflow rate and a consequent reduction of 11% in the useful volume of reservoirs due to changes in climate of the basin [16]. Designing strategies in a broader sense is extremely relevant to the formulation of actions and mechanisms to mitigate the mentioned uncertainties [17].

Water cycle fluctuations, climate change and improper land use affect several regions of the world and are key factors associated with water scarcity [5, 10, 11, 13]. In this context, the analysis of satellite data from 2012–2016 retrieved by the *Gravity Recovery and Climate Experiment (GRACE)* over eastern Brazil indicated future annual water reduction from precipitation of 16 mm, a worrying result for *EP* in a country with an energy mix predominantly composed by hydropower [13]. There is increasing doubt about how the hydropower model and water scarcity in Brazil can coexist.

In the region of the Sobradinho Reservoir, located along the lower-middle part of the São Francisco River, the following meteorological systems act [18–21] throughout the annual seasonal cycle: Intertropical Convergence Zone (*ITCZ*), South Atlantic Convergence Zone (*SACZ*); Upper Tropospheric Cyclonic Vortices (*UTCV*); remnants of frontal systems (*FS*); and the moisture convergence zone during the austral summer in *NEB*. These atmospheric systems are present mainly in the Southern Hemisphere (*SH*) summer, due to the thermodynamic characteristics of this region. The *ITCZ* moves below the Equator in March and April. The *UTCV* is present in the same period (summer) together with the *SACZ*, from which it originates, due to the air masses coming from the Amazon region and stationary *FS* coming from Southern Brazil. Therefore, given the importance of rainfall and streamflow to the maintenance of reservoirs' useful volumes, future projections indicating a decrease in these variables, threaten energy availability [2, 10, 15, 41] and impose additional challenges for the mitigation of hydrological risks. In recent years, the atmospheric systems seem to be acting less, causing less rainfall in the São Francisco River Basin.

Thus, the objective of this study was to estimate the energy generated by hydroelectric sources in *NEB* in light of climatic variables, by evaluating the criticality of water. Additionally, we aimed to: (i) describe streamflow and rainfall climatology; (ii) characterize the annual and interannual seasonality of the energy and climate variables and; (iii) define the impacts of climate variables on hydroelectric generation.

## Materials and methods

### Study area

The study area is located at the lower, middle and upper limits of the São Francisco River Basin (*SFRB*) in *NEB*, as shown in S1 Fig, encompassing Sobradinho Reservoir in the municipality of Sobradinho, Bahia [22]. Companhia Hidroelétrica do São Francisco (*CHESF*) started operating the Dam in 1979 [23], which is located 40 *km* upstream of the city of Petrolina,

Pernambuco, between longitudes 40˚5′W to 42˚W and latitudes 9˚S to 10˚S. The dam was selected for study, because it is the most important dam in *NEB*. It accounts for 58.2% of the total energy storage of the Northeast subsystem, with a mean regulated streamflow of 2,060 $m^3.s^{-1}$. Furthermore, *NEB* is the second most populated and energy-consuming region of Brazil, with a climate mainly classified as semiarid tropical.

## Data

Around the turn of the century, the Brazilian power sector experienced recurrent blackouts and energy rationing due to water shortages. In this context, data comprising the period from 2000 to 2017 were used to analyze the behavior of energy generation in *NEB*. The datasets comprised: useful volume (%) of Sobradinho Dam; hydropower generated (*MWaverage*) by the Sobradinho facility; imported energy (*MWaverage*) by the North-Northeast submarket; energy load (*MWaverage*) in *NEB*; and incremental streamflow ($m^3.s^{-1}$) at the Sobradinho gauge 168. This information is monitored by the National Electric System Operator (ONS) [4]. Incremental streamflow measured by gauge 168 ($Q_{p168}$) was calculated through Eq 1, expressing the difference between inflow ($Q_{inflow}$) in Sobradinho Reservoir and the sum of outflow ($Q_{out}$) from the Três Marias (TM) and Queimados (QM) hydropower plants, consumptive (C) use and evaporation (E) of Sobradinho Reservoir.

$$Q_{p168} = Q_{inflow} - [Q_{out}(TM + QM) + C + E] \tag{1}$$

Monthly accumulated rainfall data (*mm*) were obtained from the following weather stations of the Instituto Nacional de Meteorologia (*INMET*) for the period 2000–2017, according to S1 Table: Thus, all stations are located between the Três Marias, Queimados and Sobradinho hydroelectric plants, covering the lower, middle and upper portions of the São Francisco River Basin.

Data on monthly sea surface temperature anomaly (*SSTA*)(˚C) in the Tropical South Pacific Ocean (El Niño 3+4) were obtained from the National Center for Environment Prediction (*NCEP*) and the National Oceanic and Atmospheric Administration (*NOAA*) for the period 2000–2017 [24]. This information was used to account for oceanic indices and to identify their influence on *EP*. Furthermore, standardized Atlantic dipole data provided by *NOAA* [25] were used for statistical modeling. These data were calculated as the difference between the Tropical Northern Atlantic Index (*TNA*) measured at latitudes 5˚5′N to 23˚5′N and longitudes 15˚ W to 57˚5′W and the Tropical Southern Atlantic Index (*TSA*) measured at latitude 20˚S and longitudes 10˚E to 30˚W [26–29].

For the climatological analysis of hydrological variables, monthly rainfall (1964–2015—mm) and streamflow (1964–2017—$m^3.s^{-1}$) data were used. Accumulated rainfall data were obtained from the mentioned *INMET* stations and the Global Precipitation Climatology Project (*GPCP*) dataset [30], which provides data in a horizontal grid with 2˚5′latitude x 2˚ 5′longitude spatial resolution, developed through a combination of observational and satellite data. Incremental streamflow data at the Sobradinho gauge 168 were provided by the *ONS* and the Energy Research Company (*EPE*), part of the Ministry of Mines and Energy.

## Methodology

**Rainfall and streamflow climatology analysis.** The purpose of this analysis was to identify possible changes in incremental discharge, as expressed by Eq 1, observed at the Sobradinho gauge 168 in relation to the historical monthly series, comprising 54 years between 1964 and 2017. Descriptive statistics—mean, variance and standard deviation—were calculated for the series. Then, the hydro model is given by $G_h = F_g \cdot H \cdot \eta_h \cdot \eta_g \cdot \eta_{ge} \cdot \rho_{ag}$, where $G_h$ is

hydropower generation; $F_g$ is the gravitational constant; ($H$) is the height difference between inlet and outlet; ($\eta_g$) is the turbine efficiency; ($\eta_h$) is the hydraulic efficiency; ($\eta_{ge}$) is the energy generation efficiency; and ($\rho_{ag}$) is the density of water, which are all constants in this case. Thus, hydropower generation is directly related to streamflow. After this analysis, we tested the difference between streamflow series in the years 1964–1990 and 1991–2017, using the bilateral paired t-test (Student's t) for equal means at a significance level of $\alpha = 0.05$. The parameters mean, variance and standard deviation are presented by season. In addition, the same test was also performed for the rainfall variable, but in the periods 1964–1989 and 1990–2015.

$$D_i = Q_{t1} - Q_{t2} \ \forall \ i = 1, 2, 3, \ldots, n \tag{2}$$

where: $D_i$ is the difference between variables in the analyzed periods; $Q_{t1}$ is the variable sampled in the first period; and $Q_{t2}$ is the variable sampled in the second period.

$$s_d^2 = \frac{\sum_{i=1}^{n} (D_i - \bar{D})^2}{n - 1} \tag{3}$$

where: $s_d^2$ is the variance of the difference between variables in the distinct analyzed periods. The paired t-test statistic is given by Eq 4.

$$T = \frac{\bar{D} - \mu_i}{\frac{s_d}{\sqrt{n}}} \tag{4}$$

where: $T$ is the paired t-test statistic. This procedure was carried out in order to identify possible changes in the mean rainfall and streamflow, which are climate variables that influence power generation.

**Cluster analysis.** Additionally, cluster analysis was used with *GPCP* data to verify in which latitudes streamflow changes were more influenced by variations of rainfall, since the São Francisco River extends throughout latitudes 9˚, 10˚, 11˚, 12˚, 13˚, 14˚, 15˚, 16˚ and 17˚ South, and longitudes 42˚, 43˚, 44˚, 45˚, 46˚ and 47˚ West, comprising the entire *SFRB*. The complete linkage method was used, i.e., groups were created based on the least similar Euclidean distance values, therefore considering dissimilarities between time series in each latitude x longitude pair.

$$D = (X_1, X_2, \ldots, X_n) = max(d(X_i X_k)^2)^{\left(\frac{1}{2}\right)} \therefore \ i \neq k \tag{5}$$

where: $X_n$ is the rainfall variable.

**Mann-Kendall test.** When changes in the streamflow series were detected, we analyzed trends and seasonality of the following variables: streamflow $Q_{j=1}^n$, for Q = 1, 2, . . ., n; sea surface temperature anomaly $SSTA_{j=1}^n$, for SSTA = 1, 2, . . ., n; rainfall $Ra_{j=1}^n$, for P = 1, 2, . . ., n; useful volume of Sobradinho Reservoir $V_{j=1}^n$, for V = 1, 2, . . ., n; power generation $G_{j=1}^n$, for G = 1, 2, . . ., n; and energy imports $I_{j=1}^n$, for I = 1, 2, . . ., n. For this purpose, the Mann-Kendall test was used, to identify monotonic trends in the time series. The test statistic is given by the sign of S, expressed by:

$$S = \sum_{k=1}^{n-1} \sum_{j=k+1}^{n} sign(x_j - x_k) \tag{6}$$

where: $S > 0 = 1$, or $S = 0$, or $S < 0 = -1$. The value of z for verification of the hypothesis test

for trends is given by the following relationships:

$$z = \frac{s-1}{\sigma}, \quad \Leftrightarrow \quad S > 0 \tag{7}$$

or,

$$z = 0, \quad \Leftrightarrow \quad S = 0 \tag{8}$$

or,

$$z = \frac{s+1}{\sigma}, \quad \Leftrightarrow \quad S < 0 \tag{9}$$

In addition to the Mann-Kendall test, we also plotted the annual time series at a monthly scale.

**Principal component analysis.** Principal component analysis (*PCA*) was used to create new orthogonal datasets of the climate variables that were independent of each other. These new datasets were used for the following independent variables: (i) incremental discharge at Sobradinho gauge 168 (*Q*); (ii) useful volume of Sobradinho Reservoir (*V*); (iii) mean rainfall in the cities of the lower, middle and upper São Francisco (*Ra*); (iv) Tropical Atlantic Dipole (*Di*) and; (v) sea surface temperature anomalies in the South Tropical Pacific (*SSTA*) (*ENSO* 3 +4). These variables constitute the random vector of means $X'_{i,j}, j = \{Ra, \ Q, \ V, \ Di, \ SSTA\}$, where $X_i, j$ is the set of independent variables $i$ in month $j$. A correlation matrix (*R*) was used, due to the different scales of the original variables.

$$R_{ij} = \begin{bmatrix} r_{11} & r_{12} & \cdots & r_{1j} \\ r_{21} & r_{22} & \cdots & r_{2j} \\ \vdots & \vdots & \cdots & \vdots \\ r_{i1} & r_{i2} & \cdots & r_{ij} \end{bmatrix}$$

Diagonal of the matrix $\forall \ r_{ij} = 1$.

where: $R_{pxp}$ is a correlation matrix between independent variables. Through this matrix, it was possible to generate the eigenvalues:

$$\Lambda_{pxp} = matriz \ diagonal = \lambda_1, \lambda_2, \cdots, \lambda_n \tag{10}$$

and the eigenvectors ($e_{jp}$) which define the linear combinations according to the *PCA* model described by Eq 11. These linear combinations were used to explain power generation through a dynamic regression model by principal components.

$$\hat{y}_n = e_{j1}z_1 + e_{j2}z_2 +, \ldots, + e_{jp}z_p \tag{11}$$

where: $Y_{i,j}$ are the scores of the principal components, and $e_{jp}$ and $z_p$ are the linear combinations of the standardized vectors of the means of the original variables. The number of variables $k$ is defined according to the proportion of variance of the original variables. The new dataset (scores) calculated via *PCA* were used as exploratory variables in the dynamic regression model for the simulation of hydroelectric power generation, which is detailed in the next section.

**Dynamic regression model.** The main advantage of the dynamic regression model is that it allows the use of dependent time series, including their seasonality and trends, as independent variables. In the case of using the energy generation by hydroelectric sources, as the dependent variable, we verified its applicability regarding the lag of the series in the model.

The dynamic regression model (*DRM*) is described in Eq 12 and its parameters were estimated by the least squares method. The model allows the construction of lagged arrangements between endogenous and exogenous variables, which will produce their response after several attempts through a bottom-up process that considers the relationships between dependent and lagged explanatory variables.

$$y_{gh} = \beta_0 + y_1 y_{t-1} \quad + \ldots + y_1 y_{t-k} + \beta_{1,t} x_{1,t} +$$
$$\beta_{1,t-1} x_{1,t-1} + \cdots + \quad \beta_{1,t-k} x_{1,t-k} + \beta_{2,t} x_{2,t} + \tag{12}$$
$$\beta_{2,t-1} x_{2,t-1} + \ldots + \varepsilon_t$$

where: $Y_{gh}$ is the power generation dependent variable, $\beta_0$ is the intercept, $y_i$ is the lagged dependent variable, $\beta_n$ and $x_n$ are the coefficients and the variables used in the dynamic regression, that is, the scores of the principal components; and $\epsilon_t$ is the residual or stochastic term. In addition to the application of the dynamic model, the accuracy and precision of the results were verified by calculating *NSE*—Nash-Sutcliffe coefficient, *PBIAS*—percent bias, *MSE*—mean square error and *RMSE*—root mean square error.

## Results and discussion

The results refletc the analysis of streamflow, energy generation and energy imports, energy load, useful volume, rainfall, Atlantic dipole and *SST* anomaly in the South Pacific (*ENSO* 3+4) based on the *PCA* and dynamic regression model. Furthermore, criticality of water was verified by means of descriptive statistics and the hypothesis testing of streamflow and rainfall.

### Analysis of criticality of water

S2 Fig shows the annual distribution of incremental streamflow at Sobradinho gauge 168 in the period from 1964 to 2017, which was calculated between the Três Marias, Queimados and Sobradinho dams. Three indicators were verified: regulated streamflow, and its means in 1964–1990 and 1991–2017. Furthermore, an extreme event was observed in the beginning of the 1980s, when a strong El Niño took place, which might have contributed to increase rainfall and streamflow in the upper and middle São Francisco Basin. Similar behavior was observed in 1991–1992, characterized by a moderate El Niño and a positive dipole in the Atlantic. In addition, according to National Institute of Space Research (*INPE*) [31], there were a total of 10 occurrences of El Niño, 6 occurrences of La Niña, and 11 neutral periods from 1964 to 1990. On the other hand, 3 La Niña events, 7 El Niño events and 17 neutral periods were observed between 1991 and 2017. Improper land-use and water-use practices also increased in this second period.

S2 Fig also shows that streamflow was lower 1994 in relation to previous periods. A possible explanation is related to the increase in temperature as recently reported by the *IPCC*, which indicates aggravation of the severity of water stress associated with global warming [5, 10, 32, 33]. Energy rationing took place at the beginning of the century due to regional changes influenced by the Atlantic dipole and the anomalous positioning of the *UTCV* [18]. Streamflow in the *SFRB* is influenced by precipitation events both in *NEB* and Southeast Brazil (*SEB*), since its source is located in SEB and its mouth is located in *NEB*. Water deficit was observed in 2011–2012, 2013–2014, 2014–2015 and 2015–2016, resulting in lower accumulated rainfall, besides a gradual increase in temperature [6]. Between 2014–2015, according [8], atmospheric circulation over *SEB* experienced severe changes at the regional scale, including atmospheric blocking, which drastically reduced rainfall and negatively impacted water storage in the

basin. [3], also reported that alterations in the atmospheric circulation, detected by using a regional dynamic model, indicated a decrease in rainfall between 2012 and 2016.

The period from 1990 to 2017, stands out, due to the occurrence of more neutral periods and not as many (50% fewer) La Niña events, which can explain rainfall heterogeneity due to interannual variability and the decrease in rainfall and streamflow rates [34]. Overall, decreases in the mean, standard deviation and variance of these variables were observed in all seasons of the year, as shown in S3 Table. Such decreases were observed, when comparing the 1964–1990 and 1991–2017 periods at 5% significance level. Mean streamflow in the 1964–1990 period was $2,027 m^3 s^{-1}$, while in 1991–2017 it was $1,428 m^3 s^{-1}$ ($p–value < 0.001$). This configures a reduction of approximately 30% in streamflow ($\mu < \mu 0$). Regarding hydropower generation, it is expressed as $G_h = F_g \cdot H \cdot \eta_h \cdot \eta_g \cdot \eta_{ge} \cdot \rho_{ag}$, where $G_h$ is hydropower generation; $F_g$ is the gravitational constant; ($H$) is the height difference between inlet and outlet; ($\eta_g$) is the efficiency of the turbine; ($\eta_h$) is the hydraulic efficiency; ($\eta_{ge}$) is the efficiency of the energy generator; ($\rho_{ag}$) is the density of water, which are all constants in this case. Thus, hydropower generation is directly related to streamflow, so it also decreased in terms of mean values and presented a significant negative trend, according to the Mann-Kendall test ($\tau = -0.93$ and $p–value < 0,002$).

Hydrological changes in Sobradinho Reservoir were observed through remote sensing [1]. By using the normalized difference vegetation index (*NDVI*) and normalized difference water index (*NDWI*), the authors identified an increase in soil temperature of up to 7˚C and a reduction in surface water of up to 50%, when comparing 2015–2016 to 2011. [2], also corroborated this critical finding by reporting a reduction of 40% to 60% in the reservoir, which might be directly associated with rainfall levels, streamflow and useful volume management. In this sense, cluster analysis was carried out using precipitation data in 9 positions varying from 9˚ to 17˚ South latitude and 42˚ to 47˚25' degrees West longitude, encompassing the entire *SFRB*. The analysis indicated changes in mean rainfall patterns at latitudes 9˚, 12˚ and 13˚ South at 5% significance level. Furthermore, lower rainfall values were observed in 1990–2015, in comparison with 1964–1990 in all other latitudes. In addition, the Mann-Kendall test indicated decreasing trends in all groups. S2 Table shows the results in the cumulative column.

Rainfall time series between 1964–2015 at latitudes 9˚, 12˚ and 13˚ South encompass the cities of Paulo Afonso-BA, Sobradinho-BA, Bom Jesus da Lapa-BA and Irecê-BA, indicating a decreasing trend in precipitation by the Student t-test. The Mann-Kendall test produced the following results: latitude 9˚ ($\tau = -0.899$ and $p–value < 0.013$); latitude 12˚ ($\tau = -0.710$ and $p–value < 0.049$); which indicate negative trends. Reductions in streamflow are strongly related to meteorological conditions and land use. For example, rainfall reductions of up to 35% proportionally impacted streamflow rates [10, 35]. Thus, cluster analysis suggests reductions in rainfall of 22.9% at latitude 9˚, 13.3% at latitude 12˚ and 12.9% at latitude 13˚, according to S3 Fig with highlighting to latitude 9˚South.

Streamflow analysis presented similar behavior, with reductions in the mean streamflow value in all seasons of the 1991–2017 period compared to the 1964–1990 period. Furthermore, variance was also lower in the second period, especially during the wet season. According to S3 Table, ENSO conditions were neutral with 62% of the time in 1991–2017, while El Niño and La Niña events were 12% and 11% less frequent, respectively.

## Hydropower generation time series

S4 Fig, shows that between 2000 and 2017, the hydropower generation in *NEB* presented a decreasing trend. On the other hand, energy imports from the North submarket gradually increased, as can be seen in S5 Fig with trend revealed by the Mann-Kendall test ($\tau = 1.03$ and $p–value < 0.051$). In relation to energy load, it trended upward related to growing demand.

This energy transition is associated with operational strategies that favor importing energy from the North region, due to lower rainfall and streamflow in *NEB* [36, 37]. The behavior of hydropower generation depends, among other factors, on streamflow rates, which as previously seen in S2 Fig, drastically decreased in the period from 1991 to 2017.

In this context, one should also observe additional information on the variables: generation, imports, load, rainfall, *SSTA* in the South Pacific, Atlantic dipole and useful volume. An increase in energy imports, energy load and a substantial decrease in hydropower generation and accumulated rainfall were observed in *NEB*, during the last decade. At the seasonal scale, energy imports are more necessary during the wet season, not only due to energy demand and meteorological conditions in *NEB*, but also because it is the wet season in the northern portion of the country, which allows attenuation of the usage of hydropower storage in *NEB* [4, 34].

With the increase in energy load in *NEB*, as shown in S4 Fig, there is an urgent need to expand generation in the region, and, hence to implement alternative strategies, such as importing energy from other submarkets, due to the reduction in hydropower generation. [38] reported an increase in demand and socio-environmental conflicts related to the hydroelectric matrix [39, 40]. The authors suggested the use of mixed energy sources as a strategy to mitigate these conflicts, given the apparently irreversible persistence of this type of matrix. Energy load increases linearly and behaves quite predictably. According to data from the *ONS* [4], it increased by approximately 0.35% per year from 2000 to 2017, resulting in an average maximum load of roughly 11, 000 $MWaverage$. Currently, it is not possible to fully meet this demand using hydropower alone. Energy is imported in the most favorable rainfall and streamflow periods, coinciding with their peak values.

Rainfall time series reduced in amplitude during the 18 year period analyzed, with a negative trend estimated by the Mann-Kendall test ($\tau = -0.37$ and $p-value$ <0.030). This condition was directly influenced by streamflow rates, which also presented negative trends, as estimated by the Mann-Kendall test ($\tau = -0.44$ and $p-value$<0.009), impacting the control of Sobradinho Dam's volume and hydropower generation. [13], also reported reductions in rainfall amounts of up to 17% between 2012 and 2016 by analyzing *GRACE* data. Observational data also indicated reductions up to 22% in the same period, and 60% in 2017, coupled with a reduction of approximately 62.8% in surface water in 2017 [2]. S6 Fig shows the behavior of the useful volume time series. It represents the energy storage throughout the years, which is an independent variable controlled by the *ONS*. It reached peak values of 40% between 2000 and 2005, 80% between 2005 and 2012, and below 40% in the last five years of the series, with an overall negative trend estimated by the Mann-Kendall test ($\tau = -0.59$ and $p-value$ <0.000).

The monthly boxplot illustrated in S7 Fig describes the annual variability of rainfall. It presents the same behavior as streamflow, with critical values during winter and spring. Its distribution indicates a wet season established in the summer and autumn and a dry season established in the winter and early spring, associated with the meteorological systems that act over the region, such as the *SACZ* [21]; *UTCV* [20], *FS* [19]; and *ITCZ* [34]. Peak values occur in February and March, as a result of rainfall, consumptive uses and outflow from the Três Marias and Queimados hydropower plants. Critical periods for the operation of the electric system are in June, July, August, September and October. The useful volume for energy storage is highest in April, when it is influenced by the need to assure energy security and hydroelectric operation throughout the year, making it necessary to adopt strategies according to demand such as the use of mixed energy sources [41].

There is natural seasonality in the Atlantic dipole time series, with more positive anomalies having occurred since 2012 [26, 29]. Thus, there is an increasing trend in sea surface temperatures over the Tropical Atlantic shifting the *ITCZ* northward, reducing rainfall south of the Equator [27]. Although not strong enough to completely overcome the influence of Pacific

temperatures, it played a major role in recent water shortages observed in *NEB*, as shown by the positive Atlantic dipole trends estimated by the Mann-Kendall test ($\tau = 0.79$ and *p–value* <0.010). This behavior influences the zonal circulation of the atmosphere, particularly Walker's cell, impacting hydropower generation in *NEB* [42, 43].

## Influence of climate on energy planning

The principal components were retrieved based on the variables: energy imports, rainfall, streamflow, useful volume, Atlantic dipole and *SSTA* in the Tropical South Pacific. Through the correlation matrix shown in S4 Table, it was possible to obtain the eigenvalues, which indicated that the first three components explain 76% of the total variance of the original matrix. The estimated eigenvalues were: $\lambda_1 = 1.6$; $\lambda_2 = 1.18$ and; $\lambda_3 = 1.02$. According to Eq 11, component 1 represents mostly rainfall and streamflow, component 2 describes information related mostly to useful volume and anomalies in the *SST* in the Pacific, and component 3 represents the Atlantic dipole.

These linear combinations (loadings) originated the new dataset that was used in the analysis of energy generation, as shown in the following series of equations:

$$\hat{PC}_1 = 0.88z_1 + 0.85z_2 + 0.23z_3 + 0.13z_4 + -0.11z_5$$

$$\hat{PC}_2 = -0.11z_1 + 0.35z_2 + 0.79z_3 - 0.64z_4 + 0z_5$$

$$\hat{PC}_3 = -0.11z_1 + 0z_2 + 0.26z_3 + 0.47z_4 + 0.84z_5$$

## Influence of climate on energy generation

The following variables were used in the simulations of power generation by the dynamic regression models: rainfall, streamflow, useful volume, Atlantic dipole and *SSTA* in the South Pacific. The results of the simulations are shown in S8 Fig.

S5 Table shows the main results of the 18-year simulations by dynamic regression considering *PCA* scores. The mean square error (*MSE*) was 716, 883.1 and the root mean square error (*RMSE*) was 846.68. The model presented good fit, with residuals meeting the assumptions of normal distribution around zero, absence of outliers, independence and homoscedasticity. S8 Fig shows that the behavior of the model is similar to that of actual generation. The best fit was obtained using lagged generation data and components 1, 2 and 3 of the *PCA*, as shown in S9 Fig and S5 Table.

The estimated $\beta_0$ parameter (intercept) was 5, 750, which is approximately the mean hydropower generated in the first five years of the time series. Endogenous parameters to the energy generation model were lagged G d6 and G d13 by six and thirteen months, with coefficients of 0.28 and 0.36 for each *MW* generated. The generation trend (Trend G) with a negative sign was also considered in the independent variables. Furthermore, the exogenous coefficients originated from the principal components *PC1*, *PC2* and *PC3* were also taken into consideration, balancing the equation at the predictor $\hat{y}$.

Given the simulations of the model, its seasonality, trends, coefficient of determination and *F* statistic, the results using climate as independent variable were satisfactory. It should be mentioned that the contribution of other factors such as thermal or wind energy sources were not taken into consideration.

## Conclusion

This study identified climatological and seasonal changes in streamflow, resulting in future uncertainties regarding net hydropower generation. Streamflow analysis for the period between 1991 and 2017 revealed climatological reductions of 28% in summer, 29.4% in autumn, 31.5% in winter and 31.6% in spring. Therefore, an average reduction of 30% in streamflow was observed, which directly impacted hydropower generation. Regarding rainfall, time series analysis revealed reductions in accumulated values since 1990 of 22.9%, 13.3% and 12.9% at South latitudes 9˚ (where Sobradinho Dam is located), 12˚ and 13˚, respectively [30].

In the period from 1991 to 2017, more neutral years were observed [31], which contributed to the remarkable interannual variability due to the occurrence of seven El Niño events, three La Niña events and 17 neutral events. Increasing trends were also observed for *SSTA* in the Pacific, energy loads and energy imports, while decreasing trends were observed for hydropower generation, rainfall, streamflow and useful volume. A statistically significant decreasing trend ($p-value$ <0.05) with coefficient of determination 0.67 was found for the simulated energy generation between 2000 and 2017, as shown in S8 Fig which is crucial for modeling of hydropower generation. Therefore, we managed to incorporate climate aspects, when inferring energy generation by hydroelectric sources. Given the importance of these climate variables for energy modeling, the use of climate patterns in statistical models is recommended. In this specific case, contingency should be considered when operating the energy system, given the decreasing generation trends.

The incorporation and expansion of other energy sources, such as wind, solar, biomass and even conventional thermal plants are recommended to compensate the hydropower deficit, which was 5, 000 *MW average* in the beginning of the century and 2, 000 *MW average* in 2017, due to the gradual reduction of streamflow and precipitation, as observed in S3, S4 and S8 Figs, besides other parameters, to avoid energy rationing or blackouts like happened in 2001. Thus, based on the critical condition of water resources, there is a need for future changes in the operational strategy regarding network infrastructure. Thus, it is necessary to increase energy imports and to incorporate other sources in the energy mix to meet the growing demand observed in *NEB* and in other regions, considering variables, such as exchange of energy submarkets, population growth and industrial expansion, according to [44]. Currently, there is much debate about energy transitions and these possibilities seem to be increasingly real [39, 45]. Due to the relevance of this subject, discussions should be encouraged regarding the current uses of the dam. Multiple water uses should be considered and particularly for this region, other uses, such as human consumption and large-scale irrigation could be favored besides hydropower generation.

## Supporting information

**S1 Fig. Sobradinho Dam in Northeast Brazil.** Source: Instituto Brasileiro de Geografia e Estatística (*IBGE*). (https://portaldemapas.ibge.gov.br/portal.phphomepage).
(TIF)

**S2 Fig. Annual time series of incremental streamflow at Sobradinho gauge 168 in Northeast Brazil between 1964 and 2017.**
(TIF)

**S3 Fig. Dendrogram of rainfall at latitudes 9˚to 17˚South and longitudes 42˚to 47˚West from GPCP data 2.5˚x 2.5˚scale in Northeast Brazil between 1964 and 2015.**
(TIF)

**S4 Fig. Annual time series of generation of hydro, thermal and wind energy, besides energy load and energy imports between 2000 and 2017 in Northeast Brazil.**
(TIF)

**S5 Fig. Monthly distribution of (a) energy generation, (b) energy imports and (c) energy load in Northeast Brazil between 2000 and 2017.**
(TIF)

**S6 Fig. Annual time series of the useful volume and streamflow of Sobradinho Reservoir in Northeast Brazil between 2000 and 2017.**
(TIF)

**S7 Fig. Monthly seasonality of the variables a) precipitation, b) water streamflow and c) useful volume between 2000 and 2017 in Northeast Brazil.**
(TIF)

**S8 Fig. Simulation of energy generation by the dynamic regression model, using climate variables from *PCA* (scores—PC1, PC2 and PC3) between 2000 and 2017 in Northeast Brazil.**
(TIF)

**S9 Fig. Residual analysis of the simulations by the dynamic regression model of energy generation, using *PCA* between 2000 and 2017 in the Northeast Brazil.**
(TIF)

**S1 Table. Weather stations of INMET.**
(PDF)

**S2 Table. Descriptive statistic of rainfall in Northeast Brazil between 1964 and 2015 at latitudes 9˚ to 17˚ South and longitudes 42˚ to 47˚ West (2˚.5' x 2˚.5') with 5% significance level.** All average values are in mm.
(PDF)

**S3 Table. Descriptive statistic of incremental streamflow ($m^3.s^{-1}$) at gauge 168 of Sobradinho Dam between 1964 and 2017 in Northeast Brazil.** $Q^*$: Quarter.
(PDF)

**S4 Table. Correlation matrix $R_{ij}$ of the original variables between 2000 and 2017 in Northeast Brazil and its weights.**
(PDF)

**S5 Table. Summary of the dynamic regression model for energy generation, using the scores of the PCA between 2000 and 2017 in Northeast Brazil.**
(PDF)

**S1 Text.**
(TXT)

**S2 Text.**
(TXT)

**S3 Text.**
(TXT)

**S4 Text.**
(TXT)

**S5 Text.**
(TXT)

**S6 Text.**
(TXT)

**S1 Data.**
(XLSX)

## Author Contributions

**Conceptualization:** Nicorray de Queiroz Santos.

**Investigation:** Nicorray de Queiroz Santos.

**Methodology:** Nicorray de Queiroz Santos, Maria Helena Constantino Spyrides.

**Project administration:** Nicorray de Queiroz Santos.

**Resources:** Nicorray de Queiroz Santos, Kellen Carla Lima.

**Software:** Nicorray de Queiroz Santos.

**Supervision:** Kellen Carla Lima, Maria Helena Constantino Spyrides.

**Writing – original draft:** Nicorray de Queiroz Santos.

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
