## [Decision Letter · Decision Letter 0]

18 Dec 2020

PONE-D-20-35122

Climate and hidropower planning in the Northeast Brazil.

PLOS ONE

Dear Dr. Santos,

Thank you for submitting your manuscript to PLOS ONE. After careful consideration, we feel that it has merit but does not fully meet PLOS ONE’s publication criteria as it currently stands. Therefore, we invite you to submit a revised version of the manuscript that addresses the points raised during the review process.

We look forward to receiving your revised manuscript.

Kind regards,

Juan A. Añel, Ph.D.

Academic Editor

PLOS ONE

Journal Requirements:

3. During your revisions, please note that a simple title correction is required, 'Climate and hydropower planning in Northeast Brazil'. You may also consider making your title more specific to the study presented. Please ensure this is updated in the manuscript file and the online submission information.

4. We note that Figure 1 in your submission contain map/satellite images which may be copyrighted. All PLOS content is published under the Creative Commons Attribution License (CC BY 4.0), which means that the manuscript, images, and Supporting Information files will be freely available online, and any third party is permitted to access, download, copy, distribute, and use these materials in any way, even commercially, with proper attribution. For these reasons, we cannot publish previously copyrighted maps or satellite images created using proprietary data, such as Google software (Google Maps, Street View, and Earth). For more information, see our copyright guidelines: http://journals.plos.org/plosone/s/licenses-and-copyright.

4.1.    You may seek permission from the original copyright holder of Figure 1 to publish the content specifically under the CC BY 4.0 license. 

4.2.    If you are unable to obtain permission from the original copyright holder to publish these figures under the CC BY 4.0 license or if the copyright holder’s requirements are incompatible with the CC BY 4.0 license, please either i) remove the figure or ii) supply a replacement figure that complies with the CC BY 4.0 license. Please check copyright information on all replacement figures and update the figure caption with source information. If applicable, please specify in the figure caption text when a figure is similar but not identical to the original image and is therefore for illustrative purposes only.

Reviewers' comments:

Reviewer's Responses to Questions

**Comments to the Author**

1. Is the manuscript technically sound, and do the data support the conclusions?

Reviewer #1: Yes

2. Has the statistical analysis been performed appropriately and rigorously? 

Reviewer #1: Yes

3. Have the authors made all data underlying the findings in their manuscript fully available?

Reviewer #1: Yes

4. Is the manuscript presented in an intelligible fashion and written in standard English?

Reviewer #1: Yes

5. Review Comments to the Author

Reviewer #1: The authors analyzed the influence of climate on hydropower generation in the Sobradinho-BA dam in Brazil, during the last decades. For that, different methodologies and statistical analysis, including the evaluation of rainfall and streamflow during the last decades, as well as the simulation of hydropower generation, were developed. Authors detected a significant reduction in the hydropower generation, caused by the decrease in rainfall, and therefore, in the associated streamflow, among others. Although the manuscript has wide interest for the scientific community due to the analysis of some important implications provoked by the climate change, some parts of the manuscript need an improvement before to be published in Plos ONE. Therefore, I recommend a major revision.

1. General comment:

The manuscript is sometimes difficult to read. Overall, English needs a revision. The authors should make an effort to be more concise in some parts of the manuscript. Some examples about this are listed below:

Lines 9-14: Sentence too long. Authors should be more concise.

Lines 14-20: The text should be rephrased.

Lines 20-26: The text should be rephrased.

Lines 297-299: The text should be rephrased.

In general, writing should be revised and improved throughout the manuscript.

2. Introduction:

Lines 43-47: “In the region of the Sobradinho-BA reservoir, located at the lower-middle Sao Francisco, the following meteorological systems act [17-20] throughout the annual

seasonal cycle: the South Atlantic Convergence Zone (SACZ); upper tropospheric cyclonic vortices (UTCV); remnants of frontal systems (FS); and moisture convergence

zone during the austral summer over the NEB”. Authors should briefly explain as these meteorological systems affect the area of study.

Lines 37-38. A reference is necessary.

Lines 47-50. A reference is needed.

3. Results and discussion

Since the study is focused on the impacts of climate change, it is important that authors provide some results and projections about how the climate change will affect the area under analysis in the future. For example, in lines 313-321 authors indicate that rainfall decrease in the last years, also play a key role in the reduction of streamflow rates. In the current context of global warming, it is of crucial importance to provide some information about will be expected in the next decades. Some information about the expected future evolution of some key variables analyzed in this study, will provide a very useful information. Authors could use some local databases and related bibliography to add this information. In addition, future projections of some key variables are also available in global scale databases, as for example, the Coordinated Regional Climate Downscaling Experiment (CORDEX) or CMIP5.

Respect to the simulation of energy generation, I have some comments. Did you use a period of calibration and other period to validate the simulation? This should be specified. In addition, other parameters should be used to verify the performance of the simulation, as for example, the Nash-Sutcliffe efficiency coefficient (NSE), percent bias (PBIAS) and the ratio of the root mean square error to the standard deviation of the observed data (RSR), in order to prove the robustness of the model results.

4. Figures

Figure 1 should be improved. In the graphic presented in this image there are several mistakes that should be corrected. In addition, authors should represent the watershed in the maps.

Figure 3 caption, just below the image, is the same than in figure 2. Authors should be careful with these details.

6. PLOS authors have the option to publish the peer review history of their article (what does this mean?). If published, this will include your full peer review and any attached files.

Reviewer #1: No

---

## [Author Response · Author response to Decision Letter 0]

14 Mar 2021

Dear Reviewers,

Follow my modest review. I hope it will answer all of your answers and wishes.

Reviewer correction

 The point of reviewer: Topic 3: Title – “…title more specific to the study presented”.

Answer

New title. “The dependence of the hydropower planning in relation to the influence of Climate in the Northeast Brazil“. The change of the title intend to show a dependence between climate and hydropower generation.

Short title: Hydropower crisis and the climate.

Reviewer correction

The point of reviewer: Line 9-14: Sentence too long. Authors should be more concise.

Answer

The long sentence is due to the explanatory note of the acronym ERM – Energy Reallocation Mechanism. The idea is to consider and define how big problem of the dry period is in relation to the regulation of the electrical sector of ERM, when faced with drought. Also, because we can not to place footnote in the article (template). Anyway, we reduced.

Reviewer correction

 The point of reviewer: Line 14-20: The text should rephrased.

Answer

This is the same case as the one previously mentioned. The text put the meaning of the both ERM and PG – Physical Guarantee, in parenthesis. Both terms could be in footnotes, but it is not allowed. Anyway, we reduced.

Reviewer correction

 The point of reviewer: Line 20-26: The text should rephrased.

Answer

Considering the risks of ERM and the commitment to honor the physical guarantee, there is an inverse condition between water availability, the demand growth curve, in addition to the need to import energy and systemic restriction problems. The paragraph highlights the problem according to the context of the theme.

Reviewer correction

 The point of reviewer: Line 297-299: The text should rephrased.

Answer

The text was rephrased.

Reviewer correction

The point of reviewer: Lines 37-38: A reference is necessary.

Answer

The references were placed. 

Reviewer correction

The point of reviewer: Lines 43-47: Authors should briefly explain as these meteorological systems affect the area of study.

Answer

We made the summary about the meteorological systems between lines 49-53. 

Reviewer correction

The point of reviewer: Lines 47-50: A reference is necessary.

Answer

The references were placed. 

Reviewer correction – 3. Results and discussion

The point of reviewer: 

1. To provide some information about will be expected in next decades.

2. Authors could use some local databases and relate bibliography to add this information.

3. To use NSE, PBIAS and RSR.

Answer

1. To provide some information about the future projection, we should make dynamic numeric model of the atmosphere-ocean to capture futures scenarios, but the aim of this paper was not make a projection, but a simulations with the past dataset, using dynamic regression model. Perhaps, it will be possible in a second step of the research. 

2. This work did not aim to perform dynamic numerical simulation. Also, it did not have the character of exploring climate change.

3 . The NSE, PBIAS and RSR were incorporate into the study (Lines 202-205 and in table 4). 

Reviewer correction – 4. Figures

The point of reviewer: 

Point 4 – Figure 1: should be improved.

Point 4 – Figures: Figure 3 - About of the descriptive caption is the same of figure 2.

Answer

Figure 1 – The figure has been corrected and improved. The review figure is attached in the site.

Figure 3 - The figure caption has been corrected. The figure is attached in the site (Plos one).

Response to e-mail received on February 19, 2021 at 2:06 pm (FIGURE 1).

The clarifications regarding Figure 1:

Figure 1 was built through free software, which any researcher can use. The QGIS is an official Open Source Geospatial Foundation project. It is a free software with friendly work environment that can be used on Windows, Linux platforms, among others.

https://www.qgis.org/en_br/site/about/index.html

The database is from IBGE - Brazilian Institute of Geography. There is also no license required. The shapes (.shp files) are made available by this federal institution. From page 41 of your manual, it explains all procedures

In Brazil, there is a specific law for this as well. Decree No.8.777 of May 11, 2016 defines open data.

https://biblioteca.ibe.gov.br/visualization/livros/liv101675.pdf

The source of shapes used for construction of Figure 1 are also available on the IBGE website, which do not require a license.

https://www.ibe.gov.br/geociencias/downloads-geociencias.html

---

## [Decision Letter · Decision Letter 1]

16 Apr 2021

PONE-D-20-35122R1

The dependence of the hydropower planning in relation to the influence of Climate in the Northeast Brazil.

PLOS ONE

Dear Dr. Santos,

Thank you for submitting your manuscript to PLOS ONE. After careful consideration, we feel that it has merit but does not fully meet PLOS ONE’s publication criteria as it currently stands. Therefore, we invite you to submit a revised version of the manuscript that addresses the points raised during the review process, and my own comments (see below):

Line 10 - please, include a citation, and link if possible, to the ERM document.

l20-22: A parenthesis is missing

l35-39: This part of the text is confusing. Please, explain it better.

l42-43: The ITCZ and the SACZ are climatological features, not meteorological systems. Please, fix it.

l47: March, April.

l50-53: You should make more explicit the link here between the exposed phenomena and water availability in the region of study (or rainfall). Values of correlations, percentage of water attributable to each phenomenon, etc. In general, I miss a complete introductory discussion here on this issue.

l69: water storage? I wonder how you translate the potential data to 'energy' stored.

The data section does not contain enough information on your data sources. During the submission, you have included in the 'Data availability section links to climatological indexes. However, there is no information about getting the rainfall data from the INMET or where a reader can access it. The same happens with the incremental streamflow data (line 121). Please, all the data must be accessible and available to anyone. Include the relevant links and, if possible, the DOI of the dataset. You could upload the data to Zenodo to make them available and get a DOI.

l234, l237, l240, l259, l294 (and any other along the manuscript): Avoid using the 'surname et al.' citation style. You could use 'According to existing research [34]'

In the conclusions: It is confusing how to move from a 5000 MW deficit to 2000 MW is a worse scenario. It is necessary a better explanation here. Also, the statement on the need to increase energy imports is out of scope here. Please, remove it. This manuscript is not about the socioeconomics of the issue, and recommendations on the best strategy to deal with deficits of a power system would be a different discussion, involving studies on other possibilities, technologies, etc.

In this version, you uploaded your figures in the wrong order. Please, be more careful when you submit a reviewed version.

We look forward to receiving your revised manuscript.

Kind regards,

Juan A. Añel, Ph.D.

Academic Editor

PLOS ONE

Journal Requirements:

Reviewers' comments:

Reviewer's Responses to Questions

**Comments to the Author**

1. If the authors have adequately addressed your comments raised in a previous round of review and you feel that this manuscript is now acceptable for publication, you may indicate that here to bypass the “Comments to the Author” section, enter your conflict of interest statement in the “Confidential to Editor” section, and submit your "Accept" recommendation.

Reviewer #1: (No Response)

2. Is the manuscript technically sound, and do the data support the conclusions?

Reviewer #1: Yes

3. Has the statistical analysis been performed appropriately and rigorously? 

Reviewer #1: Yes

4. Have the authors made all data underlying the findings in their manuscript fully available?

Reviewer #1: Yes

5. Is the manuscript presented in an intelligible fashion and written in standard English?

Reviewer #1: Yes

6. Review Comments to the Author

Reviewer #1: The authors respond most of the comments, improving the quality of the manuscript. However, there are still some comments that need addressing. I recommend publication in Plos ONE if such comments are addressed.

1. Materials and Methods section

Equations corresponding to the NSE, PBIAS, MSE and RMSE parameters should be included.

2. Results and discussion section

Respect to the simulation of energy generation. Did you use a period of calibration and other period to validate the simulation? This should be specified. In addition, authors should discuss in the text the performance of the simulation based also in the results obtained for NSE and PBIAS parameters.

3.

Line 21 delete the “dash”.

4.

Line 374: delete “textit”.

7. PLOS authors have the option to publish the peer review history of their article (what does this mean?). If published, this will include your full peer review and any attached files.

Reviewer #1: No

---

## [Author Response · Author response to Decision Letter 1]

1 Jul 2021

All other answer about of the e-mail received in April 16th 2021 are in response to reviewers3.pdf.

1. Materials and Methods section

Equations corresponding to the NSE, PBIAS, MSE and RMSE parameters should be included. 

Answer of author: Due to the fact that they are classic equations and additional checks, we only commented on the use of this verification in the methodology, being commented on in the topic of dynamic regression model.

2. Results and discussion section

Respect to the simulation of energy generation. Did you use a period of calibration and other period to validate the simulation? This should be specified. In addition, authors should discuss in the text the performance of the simulation based also in the results obtained for NSE and PBIAS parameters. 

Answer of author: We considering in the table 4 the value of precision and accuracy. The table 4 considers the values of precision and accuracy. Due to the fact of the size of the article we explored this approche less.

3.

Line 21 delete the “dash”. This problem was fix.

4.

Line 374: delete “textit”. This problem was fix.

Answer in 2021, june 17th:

1) Please upload a copy of Supporting Information Figures S1-9 and Tables S1-4 which you refer to in your text on page 13-14. The figures had been uploaded. Please, could you check what might be going wrong with the upload. About to the tables, they are in the body of the text.

Answer of email - 2021-06-22.

Attached - SUPPORT INFORMATION_tables and figures.

---

## [Decision Letter · Decision Letter 2]

30 Jul 2021

PONE-D-20-35122R2

The dependence of the hydropower planning in relation to the influence of Climate in the Northeast Brazil.

PLOS ONE

Dear Dr. Santos,

Thank you for submitting your manuscript to PLOS ONE. After careful consideration, we feel that it has merit but does not fully meet PLOS ONE’s publication criteria as it currently stands. The reviewer is happy with the information that you present now, but I continue having several concerns on your work, mostly about how it is presented. Therefore, we invite you to submit a revised version of the manuscript that addresses the following points raised during the review process.

First of all, you must improve the English language use along the manuscript. Indeed, I strongly recommend that a professional service reviews the paper. I have picked a few examples of corrections necessary: The Abstract can be rewritten improving readability by a lot; some lines with awkward or wrong expressions: 46, 51, 261 (existing researchers?), 265 (this researchers), 314, 378.

Lines 88-105: This information would be better placed in a Table. Please, do it.

Table 1: You must discuss and focus in the text only on the trends statistically significative and choose a value for it (p < 0.1) should be the maximum reasonable. Under this criterium, at most, three bands seem to show significative results.

Line 320: You have mentioned GRACE before in the text without including the acronym. Therefore, you should include its explanation the first time it appears, and here (line 320), you should use only the acronym.

Line 351: Please, include evidence (references?) of this behaviour since 2012.

Lines 351-353: This statement on the link between the anomalies of the Atlantic dipole and the SST must be supported by citing relevant scientific literature.

Lines 417-419: The possibility of blackouts depends on many factors. Deficit of power production is only one of them, and others can be poor transport infrastructure, lack of interconnectivity, extreme weather, etc. However, assuming the problem of decreasing hydropower production, this would be a problem if it does not meet power demand. Here you cite a paper for Ireland to justify your statement for your region of study. This does not seem right. You should clearly state the future energy consumption scenarios for the area studied and how existing planning for 'energy security' fails by not addressing expected decreasing power production from hydropower.

We look forward to receiving your revised manuscript.

Kind regards,

Juan A. Añel

Academic Editor

PLOS ONE

Journal Requirements:

Reviewers' comments:

Reviewer's Responses to Questions

**Comments to the Author**

1. If the authors have adequately addressed your comments raised in a previous round of review and you feel that this manuscript is now acceptable for publication, you may indicate that here to bypass the “Comments to the Author” section, enter your conflict of interest statement in the “Confidential to Editor” section, and submit your "Accept" recommendation.

Reviewer #1: All comments have been addressed

2. Is the manuscript technically sound, and do the data support the conclusions?

Reviewer #1: (No Response)

3. Has the statistical analysis been performed appropriately and rigorously? 

Reviewer #1: (No Response)

4. Have the authors made all data underlying the findings in their manuscript fully available?

Reviewer #1: (No Response)

5. Is the manuscript presented in an intelligible fashion and written in standard English?

Reviewer #1: (No Response)

6. Review Comments to the Author

Reviewer #1: (No Response)

7. PLOS authors have the option to publish the peer review history of their article (what does this mean?). If published, this will include your full peer review and any attached files.

Reviewer #1: No

---

## [Author Response · Author response to Decision Letter 2]

16 Sep 2021

Dear editor, follow my comments.

English review: We request the review of writing in English for an experienced article reviewer.. The same is a Native American;

Lines 88-105: We compile the information in table format;

From(Table 1)/to(Table2): We made the comments only those that showed statistical significance T; 

Line 320: we carried out the correction, as directed;

Line 351:Such scientific references are considered;

Lines 351-353: the article shows the best correlations between the sst and the atlantic dipole.

Lines 417-419: Although, the first part of comment is not understood, as the article strictly refers to the analysis of the natural water resource, and its relationship with hydroelectric generation, we tried to adjust or associate it in the conclusion to the ten-year energy plan, in order to improve the written with the energy transition.

Answer to e-mail of September 6th.

As recommended, the reference from table 1 was inserted in the body of the text.

---

## [Editor Report · Decision Letter 3]

11 Oct 2021

PONE-D-20-35122R3The dependence of hydropower planning in relation to the influence of Climate in Northeast Brazil.PLOS ONE

Dear Dr. Santos,

Thank you for submitting your manuscript to PLOS ONE. After careful consideration, we feel that it has merit but does not fully meet PLOS ONE’s publication criteria as it currently stands. Therefore, we invite you to submit a revised version of the manuscript that addresses the points raised during the review process. Unfortunately, despite what it is claimed, the English language continues to be poor. Some parts of the manuscript are poorly written. Only in the abstract and the first line of the Introduction I have already detected several grammatical problems. For example:

Abstract:

" has cause latent perturbations"

"Energy Planning", why is it capitalized and in italics?

"latitudes 9°, 12° and 13°", North? South? East? West?

"hydro electric"  hydroelectric

"resulting each in more energy", what "each"?

Introduction

"Northeast Brazil (NEB) have experienced " "has"

Another example: In line 245 reads, "According to [1], hydrological changes in Sobradinho reservoir were observed through remote sensing". A better-written sentence would be "Hydrological changes in the Sobradinho reservoir were observed through remote sensing [1]".

Also, several of the references contain links that are broken, as those to INPE webpages.

Therefore, I ask you for a profound check of the English language and a manuscript review, including citations, references and links. Otherwise, I will have to reject your manuscript for publication.

We look forward to receiving your revised manuscript.

Kind regards,

Juan A. Añel

Academic Editor

PLOS ONE
---

## [Author Response · Author response to Decision Letter 3]

27 Oct 2021

Dear editor, 

We apologize for some basic mistakes. We did a careful review with the help of a native English speaker. We also fixed links and reference positions in the body of the text. We hope that the article is now ready for acceptance. Grateful for your attention.

---

## [Editor Report · Decision Letter 4]

2 Nov 2021

The dependence of hydropower planning in relation to the influence of Climate in Northeast Brazil.

PONE-D-20-35122R4

Dear Dr. Santos,

We’re pleased to inform you that your manuscript has been judged scientifically suitable for publication and will be formally accepted for publication once it meets all outstanding technical requirements.

Kind regards,

Juan A. Añel

Section Editor

PLOS ONE
---

## [Editor Report · Acceptance letter]

9 Nov 2021

PONE-D-20-35122R4 

The dependence of hydropower planning in relation to the influence of climate in Northeast Brazil 

Dear Dr. Santos:

I'm pleased to inform you that your manuscript has been deemed suitable for publication in PLOS ONE. Congratulations! Your manuscript is now with our production department. 

Kind regards, 

on behalf of

Dr. Juan A. Añel 

Section Editor

PLOS ONE